# Impact of Natural Graphite Flakes in Mixed Fillers on the Irradiation Behavior of Fine-Grained Isotropic Graphite

**Pengfei Lian [1,2], Heyao Zhang [3], Jinxing Cheng [4,\*], Qingbo Wang [4], Ai Yu [4], Zhao He [1,2], Jinliang Song [2,5,\*], Yantao Gao [6], Zhongfeng Tang [2,5] and Zhanjun Liu [1,2]**

1   Key Laboratory of Carbon Materials, Institute of Coal Chemistry, Chinese Academy of Sciences, Taiyuan 030001, China
2   University of Chinese Academy of Sciences, Beijing 100049, China
3   School of Microelectronics and Control Engineering, Changzhou University, Changzhou 213164, China
4   Beijing High-Tech Institute, Beijing 100094, China
5   Shanghai Institute of Applied Physics, Chinese Academy of Sciences, Shanghai 201800, China
6   School of Textiles and Fashion, Shanghai University of Engineering Science, Shanghai 201620, China
\*   Correspondence: chengjx@tsinghua.org.cn (J.C.); songjinliang@sinap.ac.cn (J.S.); Tel.: +86-13817418632 (J.S.)

**Abstract:** Two forms of fine-grained isotropic graphite, derived from mixed fillers by the isostatic pressing method, NG (filler with 100% natural graphite flake) and 75N25C-G (mixed filler with 75 wt.% natural graphite flake and 25 wt.% calcined coke) were prepared and irradiated with 7 MeV $Xe^{26+}$ to investigate its irradiation behaviors. Grazing incidence X-ray diffraction and Raman spectra show that the initial graphitization degree of 75N25C-G is lower than that of NG, but the crystallite sizes are larger due to calcined coke in the filler particles. After irradiation, the stacking height of crystallite sizes along c-axis directions ($L_c$) of NG increased, and Lc of 75N25C-G decreased. This can be attributed to irradiation-induced catalytic graphitization of calcined coke, and is also the reason that the dislocation density of 75N25C-G increases slower than that of NG. After irradiation, the crystallite sizes along a-axis directions ($L_a$) of NG and 75N25C-G reduced, but this trend was more obvious in irradiated 75N25C-G; this was closely related to the change of the surface morphology. The results show that the effect of the content of natural graphite flakes in the filler on the initial graphitization degree determines the difference in microstructure evolution caused by irradiation.

**Keywords:** fine-grained graphite; graphite flakes; irradiation

## 1. Introduction

Graphite has become one of the most extensively studied nuclear materials, starting with the first controlled and sustained critical nuclear reaction, initiated in the Chicago Pile-1 reactor. Based on the success of the graphite moderator as the core of early reactors, graphite has been used in different reactor generations, and is currently a candidate for future Generation IV high-temperature reactors (HTR) and molten salt-cooled reactors (MSR) [1]. Beyond that, graphite has also become a widely used plasma-facing component for fusion reactors [2], based its exceptional thermal conductivity, higher strength and its relatively innocuous interaction with plasma. During reactor operation, carbon atoms struck by fast neutrons are displaced from their lattice positions in the crystal structure, and a large number of interstitials and vacancies are generated [3]; this significantly changes both the graphite component dimensions and the graphite material properties [1,4]. Considering that its structural evolution, introduced by high doses of irradiation, has been confirmed to seriously influence the lifetime of the graphite used in reactors [5], there is a significant amount of neutron and ion irradiation data available on graphite, and it has always been an active research topic [6–9]. Today, in the development of two Generation IV graphite-moderated nuclear reactors, the tristructural isotropic fuel that uses pyrolitic graphite as a

key functional element has been realized in high-temperature gas-cooled reactors [10,11]; far less information is available on the irradiation behavior of candidate graphite for MSR.

MSR graphite components must minimize the impact of molten salt intrusion into the graphite pore, with very fine pore structures (<1 μm), and be able to withstand the stresses generated by the reactor environment [12,13], which is different from the uses of candidate graphite required by HTR. Most nuclear grade graphite is a mix of binder–filler material, and two typical types of porosity: gas evolution porosity that forms during volatile removal, and calcination cracks during the graphitization process, such as NBG-18, NBG-17, IG-110, ETU-10 and 2114, which increase the average diameter of the entrance pores, making many of them unsuitable for MSR application. This results from the molten salt permeation of graphite, although they exhibit good radiation stability, and their irradiation characteristics are fairly well-researched and databased [13]. For the protection of graphite from salt permeation issues, our research group has developed a series of methods to prevent molten salts successfully, including graphite with fine pores, coatings and pore impregnation [14–18]. Coatings and pore impregnation usually require a lot of time and economic cost, and cannot assure the continuous protection of graphite from salt permeation, because of interface instability and performance differences.

Fine-grained isotropic graphite (FG) may be obtained by a cold isostatic pressing method and derived from mixed fillers with natural graphite flake and calcined coke, avoiding the above negative factors. Additionally, the intrinsic, highly graphitized structure of natural graphite flake can improve the graphitization degree and increase crystallite size of obtained FG, which can contribute to improved irradiation stability [15].

In addition to the stresses generated by exposure to molten salts, of paramount interest to MSR designers is the irradiation-induced dimensional change of the graphite assemblies that is another key factor in limiting the service lifespan of the reactor core [19,20]. Irradiation-induced, highly anisotropic dimensional and property changes in graphite can lead to the generation and build-up of stresses; consequently, cracking occurs and ultimately the components fail over the course of time, challenging the structural integrity of the graphite core [4,5,21]. The mechanism of graphite irradiation-induced dimensional change, and the property changes, such as the Young's modulus, strength and thermal conductivity, are always a combination of intra- and intercrystallite effects [1–9]. Based on these results, microstructural changes in graphite, and intrinsic and irradiation-induced structural defects in nuclear graphite have been evaluated using Raman spectroscopy, which is extremely sensitive to defects. Furthermore, XRD is also a common method used to characterize irradiation-induced lattice parameter and crystal size change for understanding the behavior of graphite in nuclear reactors [6,22,23].

Because of the limitations of neutron irradiation, ion irradiation is a more attractive method, due to its higher damage rate, time efficiency, low cost and lesser residual radioactivity; ion irradiation is frequently used in the study of irradiation-induced microstructural change in graphite [24–27]. In this paper, two FGs that exhibited better molten salt barrier properties, NG (filler with 100% natural graphite flake) and 75N25C-G (75 wt.% natural graphite flake and 25 wt.% calcined coke in mixed filler) were irradiated with 7 MeV $Xe^{26+}$ to a peak damage of ∼5.0 dpa. Although high-energy ion irradiation can cause ion track formation, 7 MeV $Xe^{26+}$ irradiation only produces lattice defects by elastic collisions; owing to that, its electronic energy loss is much lower than the threshold value that may result in damaged structures along the ion track. Ion irradiation-induced microstructural changes were mainly studied using Raman spectroscopy and GIXRD, which highlights damage to the crystal lattice, and evolution of the stacking height of crystallite sizes along c-axis directions ($L_c$) and crystallite sizes along a-axis directions ($L_a$), caused by irradiation. Additionally, this was closely related to the change in surface morphology.

## 2. Materials and Methods

### 2.1. Specimen Preparation

NG and 75N25C-G were prepared in our group by using natural graphite flakes and mixed fillers of natural graphite flakes and calcined coke, respectively [15]. The properties of the NG and 75N25G-G samples are shown in Table 1.

**Table 1.** Properties of NG and 75N25G-G.

| Properties | NG | 75N25G-G |
|:---:|:---:|:---:|
| Apparent density (g/cm$^3$) | $1.85 \pm 0.02$ | $1.83 \pm 0.02$ |
| Flexure strength (MPa) | $28.6 \pm 2.5$ | $34.8 \pm 2.5$ |
| Compressive strength (MPa) | $63.5 \pm 3$ | $69.4 \pm 3$ |
| Thermal conductivity (W/m·K) | $130 \pm 2$ | $119 \pm 2$ |
| Median pore diameter (μm) | 0.183 | 0.284 |
| Open porosity (%) | $11.8 \pm 0.1$ | $13.5 \pm 0.1$ |

### 2.2. Ion Irradiation

Before irradiation, NG and 75N25C-G were cut into 5 pieces with a size of $10.0 \times 10.0 \times 2.0$ mm$^3$, polished with 1000 mesh, 2500 mesh and 5000 mesh diamond sandpaper, respectively, and then the samples were ultrasonically cleaned. The cleaned samples were observed using SEM before irradiation. The irradiation experiment was conducted at Lanzhou Institute of Modern Physics with 7 MeV Xe$^{26+}$ irradiation. The ion irradiation was then carried out at room temperature at a relatively low flux of $1 \times 10^{12}$ ions/(cm$^2$·s) to minimize any sample heating; hence, the ion beam heating effect is considered negligible [28]. The stopping and range of ions in matter (SRIM) 2010 was used to calculate the total number of atoms displaced.

Considering that the main mechanism of the irradiation effect is the displacement of atoms within the crystal structure, irradiated graphite property change should be recorded in "displacements per atom" (dpa) as a fluence. The ion irradiation processes were simulated using the "Kinchin–Pease quick calculation" mode, with a threshold displacement energy of 20 eV which was based on past experiments of ion irradiation and electron irradiation [26]. Irradiation doses of 0.1, 0.5, 2.5 and 5.0 dpa were selected to study the irradiation behavior of graphite at 0–5 dpa. According to the calculation of SRIM-2010, the surface irradiation doses are 0.02, 0.11, 0.55 and 1.25 dpa, respectively, and the irradiation depth is 2.8 μm [26]. Additionally, the corresponding irradiation fluences were $9.6 \times 10^{13}$, $4.8 \times 10^{14}$, $2.4 \times 10^{15}$ and $4.8 \times 10^{15}$ ions/cm$^2$, respectively, obtained using the conversion between the two units and the number of displacements/ion/unit depth calculated by SRIM-2010.

Like neutron irradiation, several-MeV Xe-ion irradiation produces lattice defects only by nuclear collision cascade, owing to its electronic energy losses being far below the threshold value for ion track formation. However, for heavy ion irradiation, although a high dose rate can be obtained in a short time, some inevitable disadvantages are the short penetration depth, the continuously varying dose rate over the penetration depth and the resulting surface effect. The mobility of defects allows the structure to recover at irradiation temperatures above 420 K [26], in this study, irradiation caused faster damage accumulation and amorphization at room temperature.

### 2.3. Characterizations

The samples were observed using SEM (LEO 1530VP) before irradiation, the same position was found after irradiation and the morphology was observed. GIXRD (λ = 1.2438 Å) was used to examine structural changes. The room temperature GIXRD measurements were performed at the BL14B beam line of the Shanghai Synchrotron Radiation Facility. The diffraction range was 18° < 2θ < 68°, with a step size of Δ2θ = 0.02° and a counting time of 1 s per step. The incidence angle was fixed at 0.17° to reduce the influence of

the unirradiated area of sample. The $d_{002}$ and $L_c$ were calculated by Bragg's Law and the Scherrer equation, and the graphitization degree was calculated according to formula $g = \frac{0.356 - d_{002}}{0.356 - 0.3354}$ [29].

The defects induced by irradiation were recorded using a Raman spectrometer (XploRA INV, Shanghai, China) at an excitation wavelength of 532.0 nm and effective penetration depth of about 50.0 nm. Highly ordered pyrolytic graphite (HOPG) was selected as a reference. Multiple scans were carried out at random multiple positions to avoid the influence of graphite nonuniformity on measurement results. The crystallite size was determined using the formula $L_a(\text{nm}) = 2.4 \times 10^{-10}\lambda\left(\frac{I_D}{I_G}\right)^{-1}$ [29]. The magnitude of residual stress was estimated based on the displacement of peak G: $\sigma_x(\text{MPa}) = 0.2019\Delta\omega\ (\text{cm}^{-1})$, $\sigma_y(\text{MPa}) = 0.0749\Delta\omega\ (\text{cm}^{-1})$, $\sigma(\text{MPa}) = \sqrt{\sigma_x{}^2 + \sigma_y{}^2 - \sigma_x\sigma_y}$, where $\Delta\omega$ is the Raman shift in $\text{cm}^{-1}$ and $\sigma_x$ and $\sigma_y$ are in-plane compressive residual stresses along x- and y-directions (both directions along the a-axis) in MPa. Dislocation density is given by equation $\Delta\mu^2 = K^2\ln\left(\frac{1}{b}\right)\rho(\pi\rho)^{\frac{1}{2}}$, where $\Delta\mu$ is the difference in the FWHM between the graphite sample and the reference sample and $\rho$ is the dislocation density measured in $\text{m}^{-2}$ [29]. The increase rate of defect density caused by irradiation is given by $(\rho - \rho_0)/\rho$, where $\rho_0$ is dislocation density before irradiation.

## 3. Results and Discussion

### 3.1. Morphology Changes

The SEM images of NG before and after irradiation with surface dose of 0.02 dpa (Figure 1a,b) show that some small cracks shrink after irradiation. As the dose increased to 0.11 dpa, the pore structure began to shrink. Gas evolution porosity, accommodation porosity and Mrozowski cracks generated by thermal contraction are the main components of the pore structure of graphite. Irradiation can change the geometry and size of the different types of pores found in graphite, and promote the generation of new pores. At low doses of irradiation, the c-axis expansion is accommodated by Mrozowski cracks and are presumed to eventually close with further irradiation. Since graphite irradiated at these low doses undergoes significant volume shrinkage, Mrozowski cracks make only a small contribution towards bulk dimension and density changes reported above. Most of the expansion in the c-direction is accommodated by pores of various types (gas entrapment, thermal cracks, and unfilled volumes) in graphite [30,31], thus changing the pore structure and contributing to the densification of graphite.

In addition, ridge-like structures appeared on the originally smoother surface. After irradiation at the surface irradiation dose of 0.55 dpa, the pores with smaller size shrunk obviously, and some pores even closed. Due to the increase in ridge-like structures, the surface roughness increased. Compared with the surface irradiated at 0.55 dpa, the surface irradiated by 1.25 dpa had no obvious change. The shrinkage of crack and pore structure after irradiation was similar to the morphology change of self-sintered nanopore graphite, which is due to the expansion of graphite Lc caused by irradiation and results in an increase in filler size [26]. In addition, ridge-like structures also appeared on the surface; this is caused by anisotropic swelling caused by irradiation. After irradiation at a low dose, the shrinkage of cracks and the volume shrinkage of binder—caused by catalytic graphitization—absorb the swelling of filler. However, as the dose increases, the cracks close, and irradiation-induced catalytic graphite of the binder in equilibrium with the irradiation damage. And, anisotropic swelling—represented by ridge-like structures—becomes obvious, and the pore structure is squeezed to cause shrinkage. When the irradiation dose is 0.55 dpa, the surface morphology changes, becoming saturated.

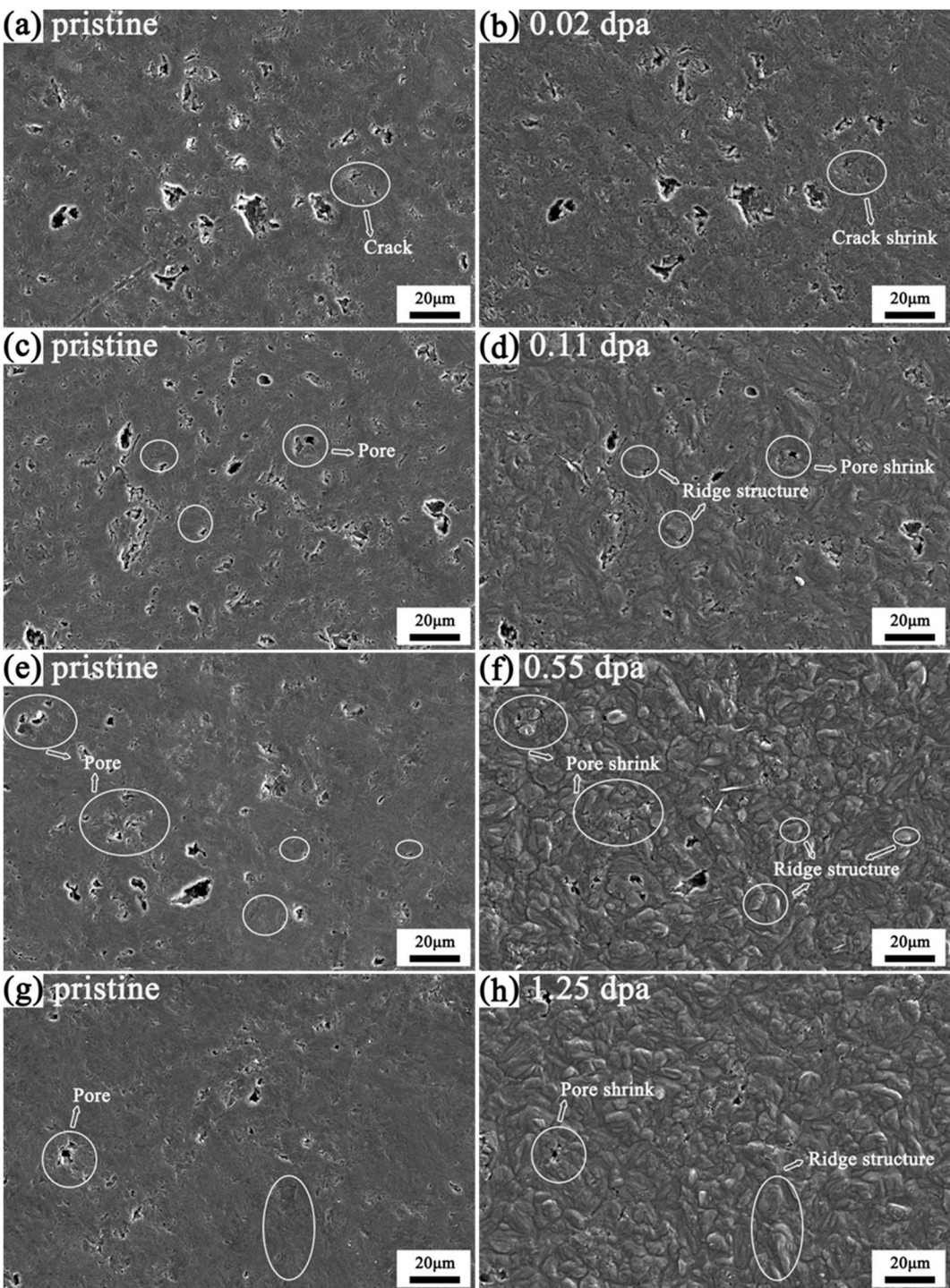

**Figure 1.** SEM images of NG before (**a**,**c**,**e**,**g**) and after irradiation at a surface irradiation dose of 0.02 dpa (**b**), 0.11 dpa (**d**), 0.55 dpa (**f**) and 1.25 dpa (**h**).

Compared with NG, the surface of 75NG25C-G before irradiation has a large number of pores; this is related to the proportion of calcined coke in the filler, as shown by Figure 2. After irradiation at 0.02 dpa, the surface of 75NG25C-G had no obvious change. With the dose increased to 0.11 dpa, cracks and pores on the surface showed obvious shrinkage, and ridge-like structures appeared on the surface. After irradiation at 0.55 dpa, some pores were closed; this is similar to the surface of NG irradiated with 0.55 dpa. However, a new crack appeared on the surface of 75NG25C-G irradiated with a dose of 1.25 dpa, and this

new crack originated from anisotropic swelling and contraction of graphite crystallites in the plane. Compared with NG, the composition of 75NG25C-G filler is more complex, so the difference in irradiation behavior of filler makes its surface morphology more sensitive to irradiation.

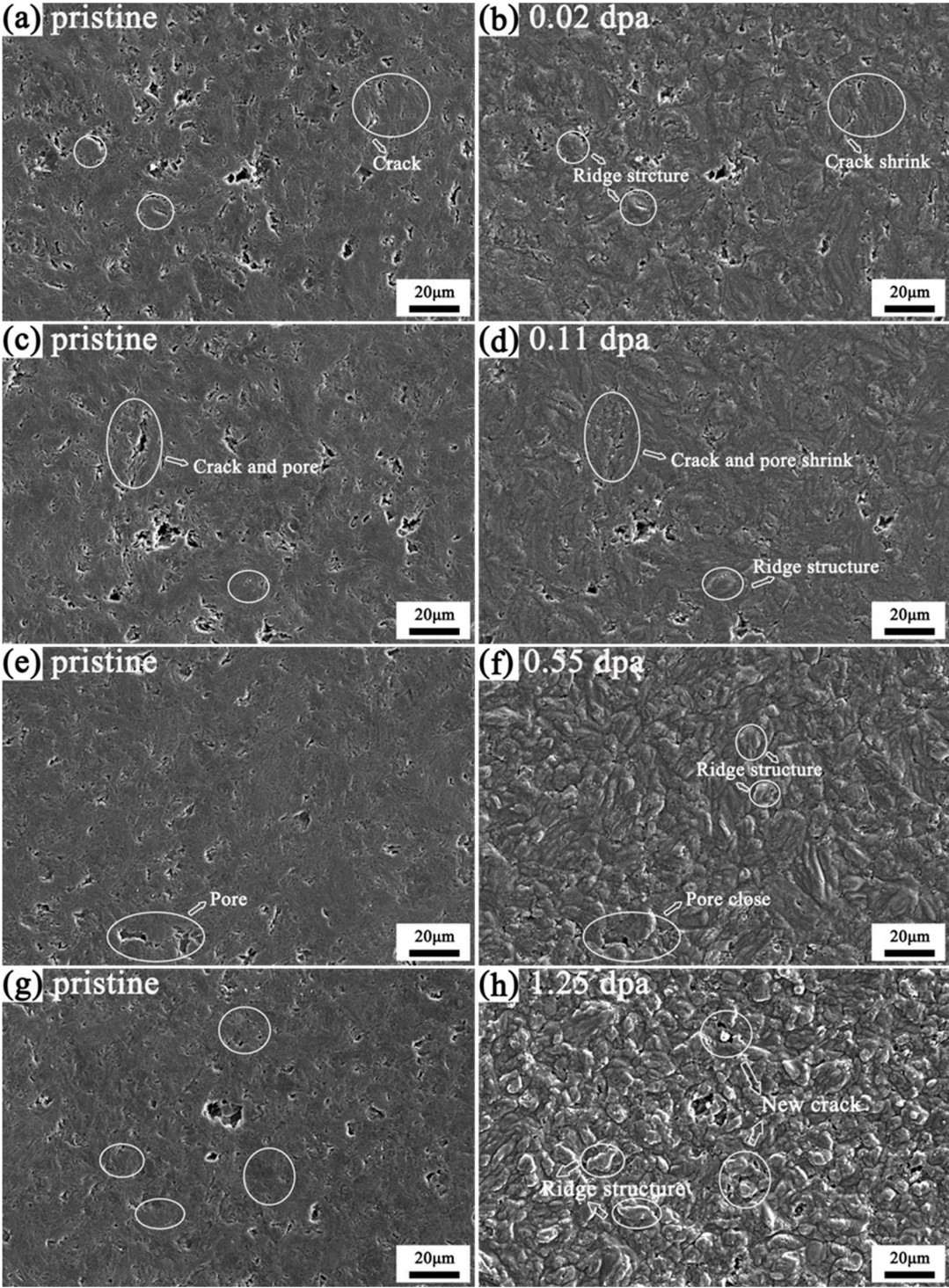

**Figure 2.** SEM images of 75N25C-G before (**a**,**c**,**e**,**g**) and after irradiation at a surface irradiation dose of 0.02 dpa (**b**), 0.11 dpa (**d**), 0.55 dpa (**f**) and 1.25 dpa (**h**).

Magnified SEM images of NG and 75NG25C-G before and after irradiation at a surface damage dose of 0.02 dpa and 1.25 dpa are shown in Figure 3. The magnified SEM

images show that after low-dose irradiation, the pores between graphite shrink due to the expansion of graphite crystallites, and the surface becomes compact and smooth. After irradiation at 1.25 dpa, the surface morphology of NG and 75NG25C-G are quite different. The surface of NG is smoother; the edges of graphite flakes and the textures formed by the accumulation of graphite sheets become blurred, and cracks shrink. However, the graphite flakes on the surface of 75NG25C-G tend to spheroidize, making the surface uneven, and as the graphite flakes spheroidize, the surface is torn and new cracks are generated.

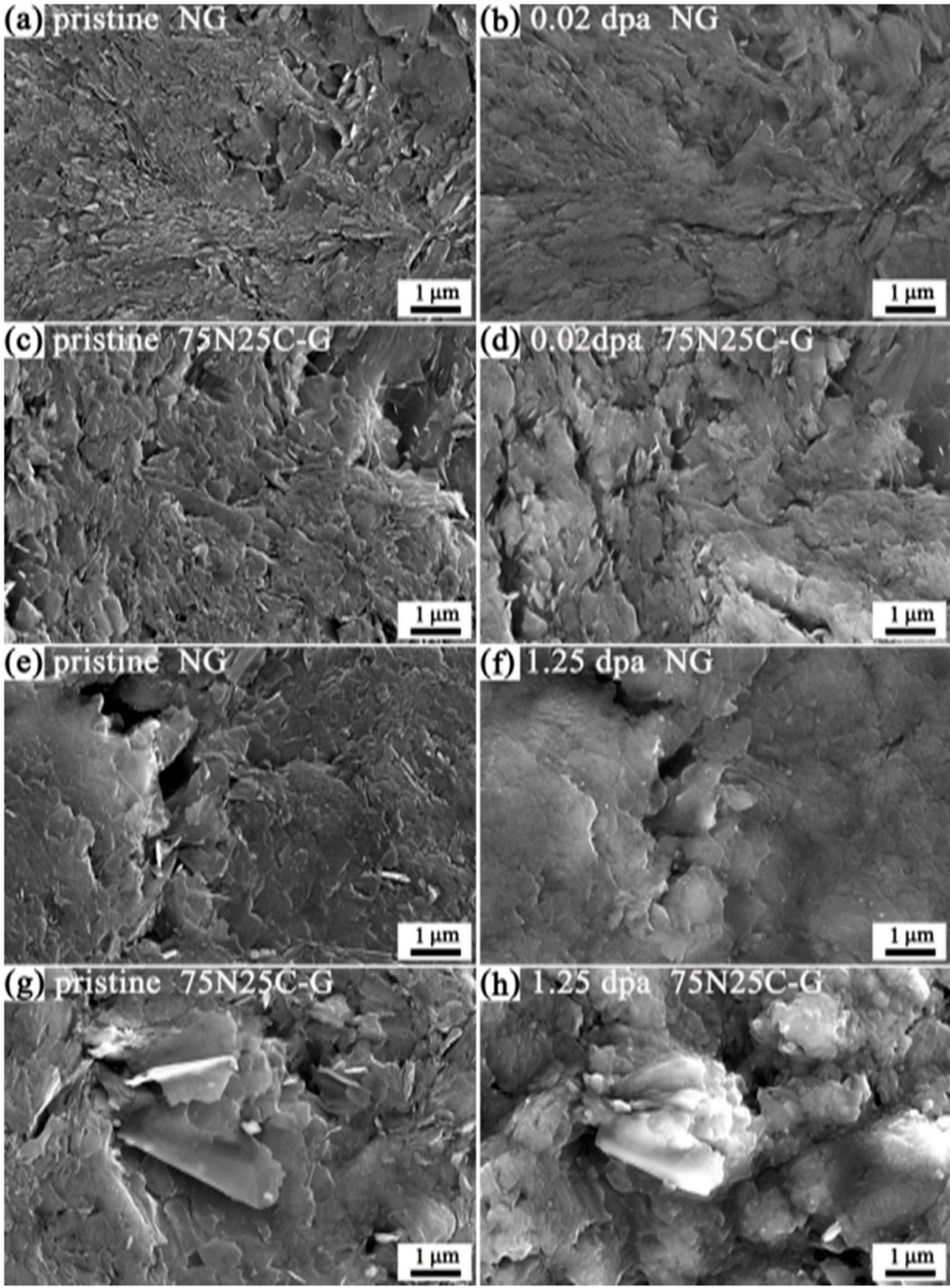

**Figure 3.** Magnified SEM images of NG and 75NG25C-G before and after irradiation at a surface damage dose of 0.02 dpa (**a–d**) and 1.25 dpa (**e–h**).

*3.2. Microstructure Variation*

In the previous research, we observed the measurement results by continuously reducing the grazing incidence angle, and found that a grazing incidence angle of 0.17° can eliminate the influence of the pristine regions as much as possible [18]. However, the measurement results are still affected by the continuous and rapid change of the damage degree with the depth in the measurement area. The irradiation evolution obtained by GIXRD is different from the surface characterization by Raman spectroscopy. Additionally, because of the small grazing incidence angle, the information obtained by GIXRD is near the surface. Therefore, the defects introduced by polishing on the surface have stronger surface effect and occupy more measurement areas for GIXRD measurement, and their graphitization degree is lower than that obtained by previous XRD measurement of unpolished graphite. GIXRD patterns of NG and 75NG25C-G are shown in Figure 4. GIXRD patterns of NG and 75NG25C-G before irradiation show that the graphitization degree of NG is slightly higher than that of 75NG25C-G, which is due to the highly ordered crystal structure of natural graphite flake. After irradiation, the (002) peaks of NG and 75NG25C-G moved to the left, indicating an increase in layer spacing and a decrease in graphitization degree, which represents the destruction of the crystal structure by radiation damage. Irradiation-induced dislocated carbon atoms cause the accumulation of a large number of interstitial atoms, formation of new basal planes and a rise in dislocations, which increases interlayer spacing and leads to microcrystals to expand along the c-axis [32,33]. Figure 4c,d show $L_c$ and the graphitization degree after different doses of irradiation. The initial stacking height of crystallites in 75NG25C-G is higher than those of NG, because the sintering ability of calcined coke is better than that of flake graphite, making the stacking height of 75NG25C-G slightly larger. After irradiation, the crystallites of NG expand along the c-axis and $L_c$ increases. However, the $L_c$ of 75NG25C-G after irradiation reduces first and then increases; this is due to the ordered stacking of disordered microcrystalline structures of calcined coke, itself due to catalytic graphitization, resulting in a decrease in $L_c$. This is confirmed by the fact that the graphitization degree of 75NG25C-G after irradiation at a low dose is slightly higher than that of NG. Figure 4c,d show that the change in the rate of $L_c$ and graphitization degree of 75NG25C-G is less than NG, which is not only related to the buffer effect of catalytic graphitization in calcined coke upon irradiation damage, but also to the larger interlayer spacing that may provide space for cascade collision. With the dose increase, the damage effect of irradiation is balanced with catalytic graphitization, and the reduction of the graphitization degree reaches saturation in a certain range.

Figure 5 shows Raman spectra of NG and 75NG25C-G filler and binder regions before and after irradiation. The D-peak in the binder region is obvious, indicating that the order of microcrystalline structure in the binder region is low. After irradiation, the width of the D-peak and G-peak increases, and the increase in relative peak intensity of the D-peak indicates the destruction of graphite planar structure by irradiation [29]; meanwhile, the change of $G'_{2D}$-peak also indicates the bending and inclination of the base plane [22,23,32]. Due to the lower irradiation temperature, the probability of vacancy–interstitial recombination is smaller, the amorphization rate shown by the Raman spectroscopy appears to be relatively quick when compared with the material irradiated with 250 keV Cl ions at an irradiation temperature above 420 K [25], and the surface area is more susceptible to the corrosion of heavy ion sputtering with higher energy [2].

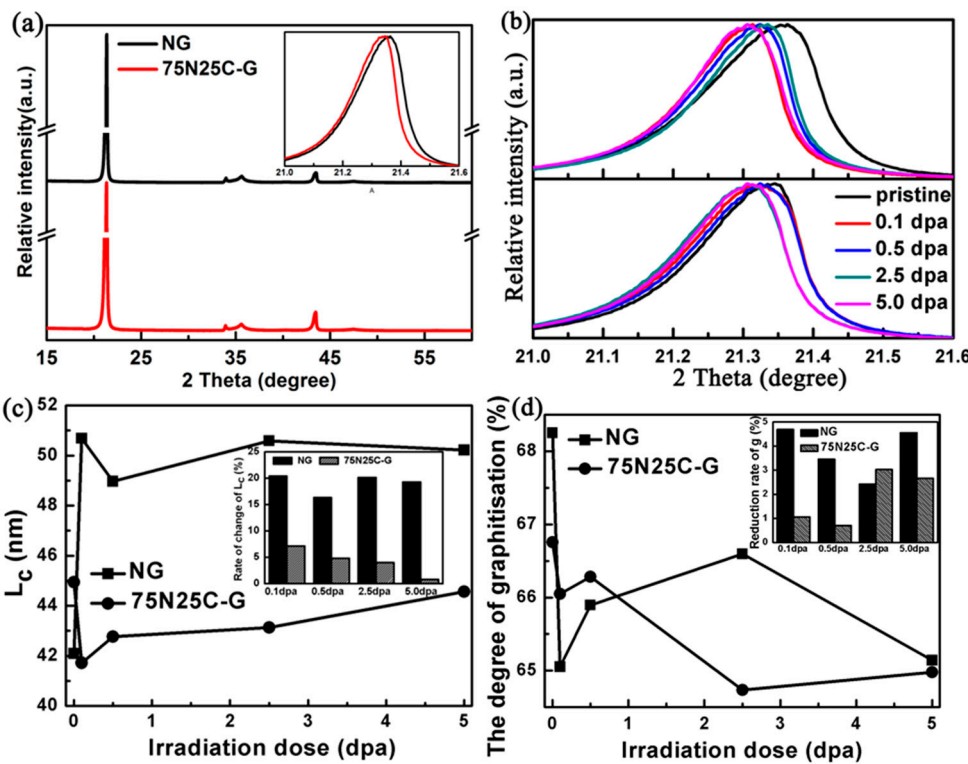

**Figure 4.** (**a**) GIXRD patterns of NG and 75NG25C-G; (**b**) (002) peak of irradiated NG and 75NG25C-G; (**c**) $L_c$ and (**d**) the degree of graphitization of NG and 75NG25C-G as functions of irradiation dose.

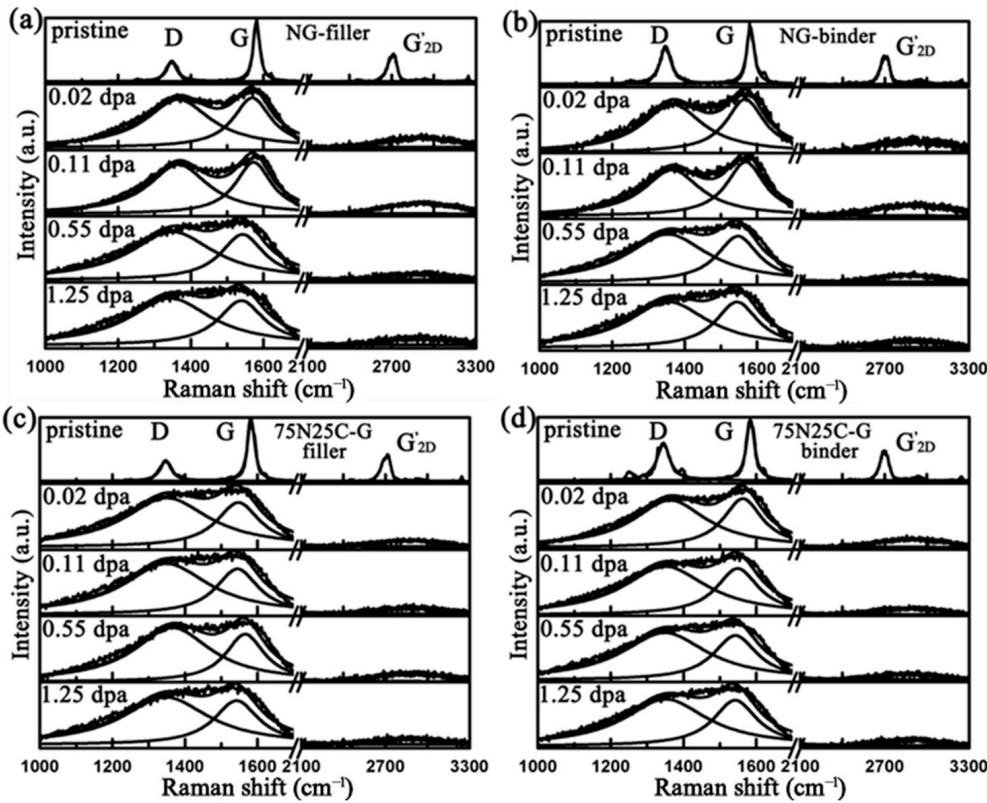

**Figure 5.** Raman spectra of the filler (**a**) and binder phase (**b**) of NG, filler (**c**) and binder phase (**d**) of 75NG25C-G.

Table 2 shows the structural characteristics of virgin and irradiated NG and 75NG25C-G graphite determined using Raman spectral analyses. The information extracted from the Raman spectrum is listed in Table 2. The initial $L_a$ of NG is slightly smaller than that of 75NG25C-G, which is similar to the results of GIXRD, and the $L_a$ of the binder region is smaller than that of the filler region. After irradiation, vacancy defects induced by irradiation form defect clusters and the crystallite size $L_a$ decreases. With the dose increase, the reduction of $L_a$ reaches saturation, and nano-crystalline graphite with a size of 9–10 nm is formed. Compared with NG, the shrinkage rate of $L_a$ in 75NG25C-G filler region and binder region irradiated with surface dose of 1.25 dpa (average 67.8%) is higher than that of NG (average 60.5%). This change in microstructure is closely related to the difference in morphology change observed using SEM. After irradiation, the surface residual stress of NG and 75NG25C-G increased from 0.09–0.38 MPa to 6.29–7.24 MPa, and there was no obvious difference in the filler region. The dislocation density increase rate in the binder region of NG and 75NG25C-G was similar, while the increase rate in the filler region of NG was larger than in the filler region of 75NG25C-G; this was due to the catalytic graphitization of calcined coke in 75NG25C-G during irradiation. Similar to the change of filler particles, the crystallite size of the binder phase region of the two kinds of graphite decrease to a larger degree; the binder phase of 75N25C-G tends to be stable at a slightly lower crystallite size, due to the influence of the initial graphitization degree. In addition, at the initial stage of irradiation, the surface stress of the binder phase is slightly larger than that of the filler particles, because the quinoline insoluble (QI) particles in the binder phase are transformed into onion-like structures, and strain is introduced in the surrounding environment. In addition, as the irradiation continues, QI particles debond and release some stress.

**Table 2.** Structural characteristics of virgin and irradiated NG and 75NG25C-G graphite determined using Raman spectral analyses.

| Graphite | | Dose (dpa) | Surface Dose (dpa) | Crystallite Size $L_a$ (nm) | In-Plane Compressive Residual Stresses (MPa) | | Average Residual Stresses (MPa) | Dislocation Density Increase Rate (%) |
|---|---|---|---|---|---|---|---|---|
| | | | | | x-Direction | y-Direction | | |
| NG | Filler particle | 0 | 0 | 35.96 | 0.11 | 0.04 | 0.09 | 0 |
| | | 0.1 | 0.02 | 11.51 | 2.26 | 0.84 | 1.98 | 1580.99 |
| | | 0.5 | 0.11 | 12.21 | 1.50 | 0.56 | 1.31 | 1493.25 |
| | | 2.5 | 0.55 | 9.42 | 7.57 | 2.81 | 6.63 | 1940.82 |
| | | 5.0 | 1.25 | 10.18 | 8.27 | 3.07 | 7.24 | 2084.26 |
| | Binder phase | 0 | 0 | 19.94 | 0.32 | 0.12 | 0.28 | 0 |
| | | 0.1 | 0.02 | 13.24 | 2.61 | 0.97 | 2.28 | 1099.89 |
| | | 0.5 | 0.11 | 15.05 | 2.38 | 0.88 | 2.08 | 1097.27 |
| | | 2.5 | 0.55 | 9.51 | 6.65 | 2.47 | 5.82 | 1272.72 |
| | | 5.0 | 1.25 | 10.11 | 7.19 | 2.67 | 6.29 | 1294.09 |
| 75N25C-G | Filler particle | 0 | 0 | 37.74 | 0.44 | 0.16 | 0.38 | 0 |
| | | 0.1 | 0.02 | 8.52 | 2.87 | 1.07 | 2.52 | 1205.34 |
| | | 0.5 | 0.11 | 8.81 | 6.67 | 2.47 | 5.84 | 1399.75 |
| | | 2.5 | 0.55 | 8.56 | 7.19 | 2.67 | 6.30 | 1376.98 |
| | | 5.0 | 1.25 | 9.19 | 7.71 | 2.86 | 6.75 | 1459.24 |
| | Binder phase | 0 | 0 | 24.60 | 0.32 | 0.12 | 0.28 | 0 |
| | | 0.1 | 0.02 | 10.56 | 3.44 | 1.28 | 3.02 | 940.21 |
| | | 0.5 | 0.11 | 9.29 | 6.30 | 2.34 | 5.52 | 1057.65 |
| | | 2.5 | 0.55 | 9.36 | 7.58 | 2.81 | 6.64 | 1120.54 |
| | | 5.0 | 1.25 | 9.87 | 7.65 | 2.84 | 6.70 | 1102.75 |

## 4. Conclusions

The irradiation behavior of NG and 75N25C-G was studied by simulating neutron irradiation damage with 7 MeV $Xe^{26+}$ irradiation. NG and 75N25C-G showed similar

changes in morphology after irradiation with a surface dose of 0–0.55 dpa. When graphite flakes are more tightly stacked, the surface cracks and pores shrink. However, after irradiation at surface dose of 1.25 dpa, new cracks appear on the surface of 75N25C-G due to the spheroidization of the graphite flakes on the surface. The graphitization degree of NG is slightly higher than that of 75N25C-G; however, the initial $L_c$ of 75N25C-G is larger, which is due to calcined coke in the filler. After irradiation, the $L_c$ of NG expands due to the distortion of interstitial atoms and lattice planes caused by irradiation, and the graphitization degree decreases. The $L_c$ of 75N25C-G decreases first and then increases; the degree of graphitization decreases less than that of NG, which indicates that the irradiation-induced catalytic graphitization appears in the calcined coke of 75N25C-G filler. The initial $L_a$ of 75N25C-G is larger, and the dislocation density increases slowly; however, the shrinkage of $L_a$ is more significant, which is related to the phenomena observed using SEM.

To sum up, the study of irradiation behavior shows the competitive or synergistic effects of irradiation-induced catalytic graphitization on irradiation damage production. In NG, the weight ratio of natural graphite flakes in the filler affects the change of the crystallite size of graphite through the determined initial graphitization degree, and then produces different morphological changes and cracks on the surface.

**Author Contributions:** Conceptualization, H.Z. and J.C.; methodology, P.L. and Q.W.; software, A.Y.; validation, H.Z., J.C. and Z.H.; formal analysis, J.S.; investigation, Y.G.; resources, J.S.; data curation, J.C.; writing—original draft preparation, P.L.; writing—review and editing, H.Z.; visualization, Z.T.; supervision, J.S.; project administration, Z.L.; funding acquisition, J.S. All authors have read and agreed to the published version of the manuscript.

**Funding:** This work was financially supported by the National Natural Science Foundation of China (No. 52072397, No. 12175323, No. 12005289); Institute of Coal Chemistry, Chinese Academy of Sciences (No. SCJC-XCL202209); Supported by the DNL Cooperation Fund, CAS (DNL202012).

**Institutional Review Board Statement:** Not applicable.

**Informed Consent Statement:** Not applicable.

**Data Availability Statement:** Not applicable.

**Conflicts of Interest:** The authors declare no conflict of interest.

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
