# Peer review of "Impact of Natural Graphite Flakes in Mixed Fillers on the Irradiation Behavior of Fine-Grained Isotropic Graphite"

_crystals, doi:10.3390/cryst12121819_

Round 1
Reviewer 1 Report
Several aspects of the manuscript need to be improved to accept this article.
General corrections
Table 1 – The headings of this table are missing
Page 3 – The sentence in row 115 is not complete. The context of this sentence is crucial for the dpa calculations.
General comments
Section 2.2
Please provide the ion fluence and displacement energy. This information should always be provided to verify and compare the SRIM calculations with other research.
Discussion
The discussion needs to expand upon the limitations of this research.
Please add a sentence or two explaining the differences between ion implantations and neutron irradiation. This context is needed to understand that the Raman and XRD measurements have an extra layer of complexity when studying ion-irradiated surfaces.
XRD results in the discussion sections need to be improved. The range of penetration of XRD is much larger than Raman spectroscopy. In this case, the XRD measurements will be affected by the polishing of the surface (introduction of damage), ion implantation, and pristine regions.
The ion implantations were conducted close to room temperature, probably leading to an accelerated amorphization of the material.
The amorphization rate of the material shown by the Raman spectroscopy appears to be relatively quick when compared to other research (Ammar, M.R., et al., N. Characterizing various types of defects in nuclear graphite using Raman scattering: heat treatment, ion irradiation, and polishing. Carbon 2015, 95, 364-373). Please check your dose calculations or explain the role of low temperature in your experiments.
Neutron irradiation effects typically do not affect gas evolution pores (round pores) in nuclear graphite. Context about how the different types of pores are affected by neutron irradiation needs to be provided to not mislead the reader. Gas evolution porosity, unfilled pores, calcination cracks, and Mrozowski cracks would behave differently under neutron irradiation or ion implantation. Some of these pores would not be as affected by irradiation. In this manuscript, a relatively low dose significantly affects all the pores.
Author Response
General corrections
Point 1:
“Table 1 – The headings of this table are missing”
“Page 3 – The sentence in row 115 is not complete. The context of this sentence is crucial for the dpa calculations.”
Response 1: Thanks for the reviewer’s comment and we are sorry for the unclear text. We carefully checked the manuscript and revised it.
General comments
Point 2:
-Section 2.2-
Please provide the ion fluence and displacement energy. This information should always be provided to verify and compare the SRIM calculations with other research.
Response 2: Thanks for the reviewer’s comment. It’s necessary to indicate the the ion fluence and displacement energy, and this obviously increases the readability of this paper. The fluence (ion/cm2) and displacement energy have been added to the manuscript as following:
Page3, Line119-122: “The ion irradiation processes were simulated by the “Kinchin-Pease quick calculation” mode with threshold displacement energies of 20 eV, which was based on past experiments of ion irradiation and electron irradiation [26]. ”
Page3, Line125-127: “And, the corresponding irradiation fluences were 9.6 × 1013, 4.8 × 1014, 2.4 × 1015 and 4.8 × 1015 ions/cm2, respectively, obtain by the conversion between the two units and the number of displacements/ion/unit depth calculated by SRIM-2010.”
Point 3:
-Discussion-
The discussion needs to expand upon the limitations of this research.
Please add a sentence or two explaining the differences between ion implantations and neutron irradiation. This context is needed to understand that the Raman and XRD measurements have an extra layer of complexity when studying ion-irradiated surfaces.
Response 3: Thanks for reviewer’s comment and is very helpful to improve the paper. The differences between ion implantations and neutron irradiation have been added to the manuscript as following:
Page3, Line128-133: “Similar with neutron irradiation, several-MeV Xe-ion irradiation produces lattice defects only by nuclear collision cascade, owing to that it’s electronic energy losses far below the threshold value for ion track formation. However, for heavy ion irradiation, although a high dose rate can be obtained in a short time, the inevitable drawback disadvantage is the short penetration depth and the continuously varying dose rate over the penetration depth, and the resulting surface effect. ”
Point 4
XRD results in the discussion sections need to be improved. The range of penetration of XRD is much larger than Raman spectroscopy. In this case, the XRD measurements will be affected by the polishing of the surface (introduction of damage), ion implantation, and pristine regions.
Response 4: Thanks for reviewer’s comment. More details of have been added as follows:
Page5, Line221-231: ‘‘In the previous research, we observed the measurement results by continuously reducing the grazing incidence angle, and found that a grazing incidence angle of 0.17o can eliminate the influence of the pristine regions as much as possible [18]. However, the measurement results are still affected by the continuous and rapid change of the damage degree with the depth in the measurement area. The irradiation evolution obtained by GIXRD is different from the surface characterization by Raman spectroscopy. And, because of the small grazing incidence angle, the information obtained by GIXRD is near the surface. Therefore, the defects introduced by polishing on the surface have stronger surface effect and occupy more measurement areas for GIXRD measurement, and their graphitization degree is lower than that obtained by previous XRD measurement for unpolished graphite.’’
Point 5
The ion implantations were conducted close to room temperature, probably leading to an accelerated amorphization of the material.
Response 5: Thank the reviewer’s professional comments. We did forget to consider the influence of irradiation temperature on irradiation effect, although this is very important. For this, we have added the following:
Page3, Line133-136: ‘‘Considering that the mobility of defects allows the structure to recover at irradiation temperature above 420 K[26], the faster damage accumulation and amorphization obtain by irradiation at room temperature in this study.’’
Point 6
The amorphization rate of the material shown by the Raman spectroscopy appears to be relatively quick when compared to other research (Ammar, M.R., et al., N. Characterizing various types of defects in nuclear graphite using Raman scattering: heat treatment, ion irradiation, and polishing. Carbon 2015, 95, 364-373). Please check your dose calculations or explain the role of low temperature in your experiments.
Response 6: Thank the reviewer’s comments and the explanation of this have been added to the manuscript as following:
Page10, Line264-269: ‘‘Due to the lower irradiation temperature, the probability of vacancy-interstitial recombination is smaller, amorphization rate shown by the Raman spectroscopy appears to be relatively quick when compared to the material irradiated with 250 keV Cl ions at irradiation temperature above 420 K [25], and the surface area is more susceptible to the corrosion of heavy ion sputtering with higher energy[2].’’
Point 7
Neutron irradiation effects typically do not affect gas evolution pores (round pores) in nuclear graphite. Context about how the different types of pores are affected by neutron irradiation needs to be provided to not mislead the reader. Gas evolution porosity, unfilled pores, calcination cracks, and Mrozowski cracks would behave differently under neutron irradiation or ion implantation. Some of these pores would not be as affected by irradiation. In this manuscript, a relatively low dose significantly affects all the pores.
Response 7: We sincerely appreciate the valuable comments that will help us improve it to a better scientific level. According to reviewer’s suggestion, the manuscript is revised as follows
Page4, Line164-174:‘’Gas evolution porosity, accommodation porosity or Mrozowski cracks generated by thermal contraction are the main components of the pore structure of graphite. Irradiation can change the geometry and size of the different types of pores found in graphite and promote the generation of new pores. At low doses of irradiation, the c-axis expansion is accommodated by Mrozowski cracks and are presumed to eventually close with further irradiation. Since graphite irradiated at these low doses undergoes significant volume shrinkage, Mrozowski cracks have only a small contribution towards bulk dimension and density changes reported above. Most of the expansion in the c-direction is accommodated by pores of various types (gas entrapment, thermal cracks, and unfilled volumes) in graphite[30,31], thus changing the pore structure and contributing to the densification of graphite.
Ref.
[30] Arregui-Mena, J.D.; Cullen, D.A.; Worth, R.N.; Venkatakrishnan, S.V.; Jordan, M.S.L.; Ward, M.; Parish, C.M.; Gallego, N.; Katoh, Y.; Edmondson, P. D.; Tzelepi, N. Electron tomography of unirradiated and irradiated nuclear graphite. J. Nucl. Mater. 2020, 545, 152649.
[31]Contescu, C.I.; Arregui-Mena, J.D.; Campbell, A.A.; Edmondson, P.D.; Gallego, N.C.; Takizawa, K.; Katoh, Y. Development of mesopores in superfine grain graphite neutron-irradiated at high fluence. Carbon 2019, 141:663-675.

Reviewer 2 Report
The article presents a large amount of experimental data on the effect of irradiation on the morphology and fine structure of graphite.
In section 2.1 (Specimen preparation), the authors indicate that graphite samples were obtained by the method described in [He, Z.; Lian, P.F.; Song, J.L.; Zhang, D.Q.; Liu, Z.J.; Guo, Q.G. Microstructure and properties of fine-grained isotropic graphite based on mixed fillers for application in molten salt breeder reactor. J. Nucl. Mater. 2018, 511, 318-327]. In that work, the sintering temperature of the samples was 3273 K and the degree of graphitization was more than 90%. Therefore, it is necessary to give in the text several suggestions on the technology of sample preparation and explain why in this case the initial degree of graphitization is less than 70%.
It is also desirable to indicate that the Raman spectra were obtained both from filler particles and from binder phases and briefly discuss the results (presented in Table 2) of changes in the binder phases.
Author Response
Point 1
The article presents a large amount of experimental data on the effect of irradiation on the morphology and fine structure of graphite.
In section 2.1 (Specimen preparation), the authors indicate that graphite samples were obtained by the method described in [He, Z.; Lian, P.F.; Song, J.L.; Zhang, D.Q.; Liu, Z.J.; Guo, Q.G. Microstructure and properties of fine-grained isotropic graphite based on mixed fillers for application in molten salt breeder reactor. J. Nucl. Mater. 2018, 511, 318-327]. In that work, the sintering temperature of the samples was 3273 K and the degree of graphitization was more than 90%. Therefore, it is necessary to give in the text several suggestions on the technology of sample preparation and explain why in this case the initial degree of graphitization is less than 70%.
Response 1: Thanks for reviewer’s comment and it is very important to improve the manuscript. The explanation of this have been added to the manuscript as following:
Page8, Line226-231:“And, Because of the small grazing incidence angle, the information obtained by GIXRD is near the surface. Therefore, the defects introduced by polishing on the surface have stronger surface effect and occupy more measurement areas for GIXRD measurement, and their graphitization degree is lower than that obtained by previous XRD measurement for unpolished graphite.”
Point 2
It is also desirable to indicate that the Raman spectra were obtained both from filler particles and from binder phases and briefly discuss the results (presented in Table 2) of changes in the binder phases.
Response 2: Thanks for reviewer’s comment. We think this is an excellent suggestion and have added a brief discussion to the revised manuscript.
Page10, Lin288-295 ‘‘Similar to the change of filler particles, the crystallite size of the binder phase region of the two kinds of graphite shrinkage at a larger degree, and the binder phase of 75N25C-G tends to be stable at a slightly lower crystallite size due to the influence of initial graphitization degree. In addition, at the initial stage of irradiation, the surface stress of the binder phase is slightly larger than that of the filler particles, because the quinoline insoluble (QI) particles in the binder phase are transformed into onion-like structures and strain is introduced in the surrounding environment. In addition, as the irradiation continues, QI particles debond and release some stress.’’

Round 2
Reviewer 1 Report
This new version of the paper is acceptable.